# CONTEXT-AWARE PROMPT TUNING: ADVANCING IN-CONTEXT LEARNING WITH ADVERSARIAL METHODS

## ABSTRACT

Large Language Models (LLMs) can perform few-shot learning using either optimization-based approaches or In-Context Learning (ICL). Optimization-based methods often suffer from overfitting, as they require updating a large number of parameters with limited data. In contrast, ICL avoids overfitting but typically underperforms compared to optimization-based methods and is highly sensitive to the selection, order, and format of demonstration examples. To overcome these challenges, we introduce Context-aware Prompt Tuning (CPT), a method inspired by ICL, Prompt Tuning (PT), and adversarial attacks. CPT builds on the ICL strategy of concatenating examples before the input, extending it by incorporating PT-like learning to refine the context embedding through iterative optimization, extracting deeper insights from the training examples. Our approach carefully modifies specific context tokens, considering the unique structure of the examples within the context. In addition to updating the context with PT-like optimization, CPT draws inspiration from adversarial attacks, adjusting the input based on the labels present in the context while preserving the inherent value of the user-provided data. To ensure robustness and stability during optimization, we employ a projected gradient descent algorithm, constraining token embeddings to remain close to their original values and safeguarding the quality of the context. Our method has demonstrated superior accuracy across multiple classification tasks using various LLM models, outperforming existing baselines and effectively addressing the overfitting challenge in few-shot learning.

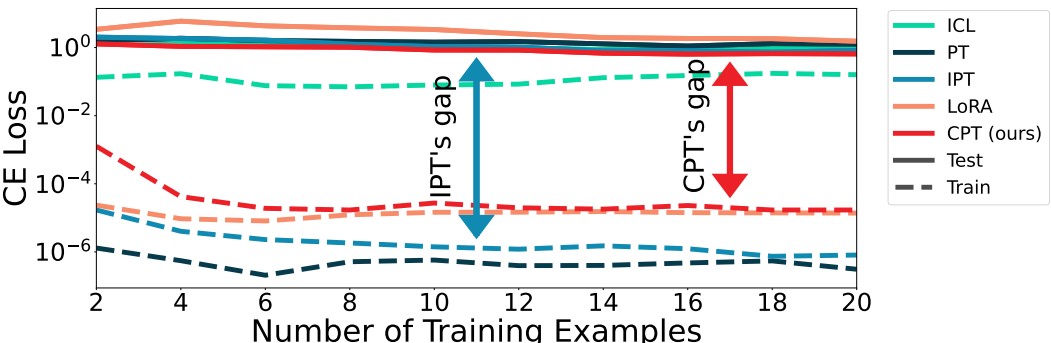

Figure 1: **Overfitting Comparison: CPT vs. Baselines** Visualizing the train-test loss gap across various methods and training set sizes using the GPT-j model on the DBpedia dataset. For each model, there are two loss graphs: one for train loss (dotted line) and one for test loss (solid line). CPT performs better in mitigating overfitting compared to optimization-based methods. Despite a relatively higher training loss, CPT achieves the lowest test loss.

[†]Department of Computer Science Technion - Israel Institute of Technology
[‡]School of Electrical and Computer Engineering - Ben-Gurion University of the Negev

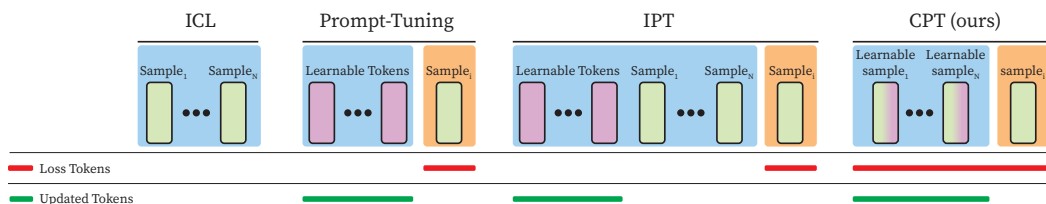

Figure 2: **Comparison of Baseline Algorithms Training and Token Utilization.** We highlight the key differences between CPT and the baselines, focusing on ICL, PT, and IPT. For each method, we emphasize two types of tokens: those used as a prefix to the input (blue background) and those used for loss (orange background). In addition, we split the prefix tokens into two groups: those updated via the training process, called *Learnable tokens* (pink), and those remaining fixed during the training process, called *Sample tokens* (green). CPT features *Sample tokens* in dual colors, reflecting their progression from *Sample tokens* to *Learnable tokens* as they are optimized.

# 1 INTRODUCTION

Fine-tuning Large Language Models (LLMs) is a widely used technique that adapts models to specific tasks by modifying all their parameters. Despite its effectiveness, this approach requires handling billions of parameters, which can be prohibitively expensive and inefficient, especially in terms of computational resources and storage, making it challenging to scale effectively.

To address the limitations of fine-tuning, several parameter-efficient methods have been introduced. Low-Rank Adaptation (LoRA) (Hu et al., 2021) reduces the number of trainable parameters by learning a low-rank decomposition. However, it still requires a portion of the model's weights, which remains burdensome, particularly since state-of-the-art models like Llama3 (AI@Meta, 2024) and GPT-4 (OpenAI, 2024) typically range from 7 billion to 1.7 trillion parameters. Another approach, Prompt Tuning (PT) (Lester et al., 2021), offers a more efficient solution by updating a small set of learnable token embeddings, which are concatenated before the input, while leaving the LLM's weights completely untouched. Alternatively, In-Context Learning (ICL) (Brown et al., 2020) adjusts the model to new tasks without any parameter updates, relying on the straightforward concatenation of training examples with the input context. Despite its computational efficiency, recent studies Zhang et al. (2022); Sun et al. (2023); Perez et al. (2021) indicate that ICL falls short compared to supervised fine-tuning methods. To leverage the strengths of both PT and ICL, Instruction Prompt Tuning (IPT) (Singhal et al., 2022) was introduced. This approach involves concatenating both learnable tokens and context to the input, training only the learnable tokens while keeping the context and model weights frozen.

Despite the recent advancements in parameter-efficient methods, determining the optimal method for few-shot learning remains highly unsettled. On one hand, optimization-based methods such as fine-tuning, LoRA, PT, and IPT are prone to overfitting, especially in few-shot settings where the number of trainable parameters is large – a condition known to exacerbate overfitting, as demonstrated in fig. 1. Meanwhile, In-Context Learning (ICL) mitigates overfitting by avoiding model parameter updates; however, it does not match the performance of other methods. Consequently, the most effective method for various different scenarios remains uncertain (Sun et al., 2023).

Context-aware Prompt Tuning (CPT), fuses concepts from In-Context Learning (ICL), Prompt Tuning (PT), and adversarial attacks (Blau et al., 2022; 2023; Carlini & Wagner, 2017; Athalye et al., 2018; Biggio et al., 2013; Goodfellow et al., 2014; Kurakin et al., 2016; Nguyen et al., 2015; Madry et al., 2017; Rebuffi et al., 2021; Gowal et al., 2020) into a cohesive approach, with the main differences from the baselines illustrated in fig. 2. CPT follows the ICL technique of concatenating training examples prior to the input. Similarly to PT, CPT updates only the context token embeddings through iterative optimization, leveraging again the training examples present in the context. However, CPT carefully refines the context tokens while accounting for the context's unique structure, keeping the label tokens intact, preserving their role as the ground truth. To effectively reduce overfitting and enhance performance, CPT adopts two strategies inspired by adversarial attacks: in-

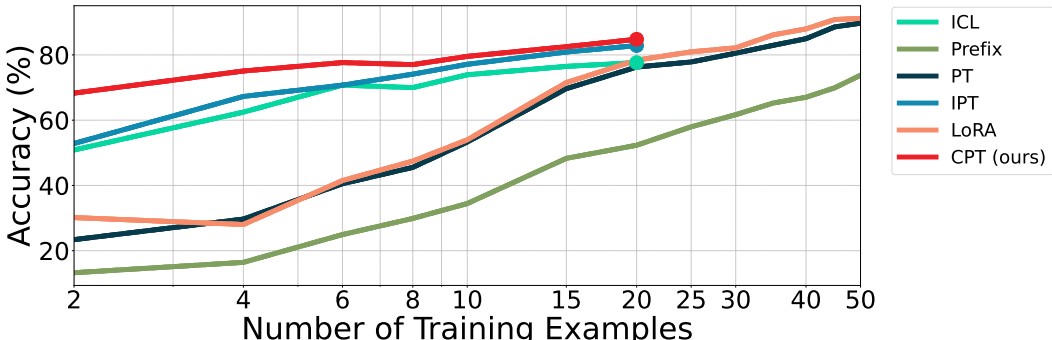

Figure 3: **Few-Shot Methods Comparison.** We compare CPT using the GPT-j model and the DBpedia dataset to baselines in few-shot settings, showing that it particularly excels when dealing with a limited number of examples. Additionally, we show that context-based methods hit memory constraints (marked with a dot) as the number of training examples rises beyond a certain level.

corporating context labels into the loss function and applying projection after updating the token embeddings. By including context labels in the loss, CPT refines input adjustments, guiding the model to optimize the entire context rather than focusing solely on the training label. To further mitigate overfitting, projected gradient descent is applied after each optimization step. This method ensures that token embedding updates remain within a controlled range, preserving proximity to their original values, under the assumption that user-provided examples are valuable. Additionally, CPT employs a loss weighting approach leverages recency bias – a phenomenon highlighted by Zhao et al. (2021), where the model tends to prioritize examples located nearer the end of the context. We recommend leveraging this property by applying an exponentially decaying weight to examples as they approach the beginning of the context, thereby increasing the emphasis on more recent examples in the optimization process.

We validate our method through a comprehensive evaluation of several classification tasks and include extensive ablations. We demonstrate that CPT outperforms other baselines across nearly every scenario, as shown in fig. 3. We use diverse templates and seeds, which is crucial due to ICL's sensitivity to training examples and format selection, as highlighted by Sun et al. (2023); Zhao et al. (2021).

To summarize, our key contributions are as follows:

- We propose a novel few-shot method called Context-aware Prompt Tuning that enhances ICL with PT and adversarial methodologies. Our method carefully optimizes the context tokens while accounting for their unique structure.
- Our method incorporates ground truth labels from the context into the loss term, optimizes with projected gradient descent, and applies recency-bias-inspired loss weighting.
- We achieve state-of-the-art results on several classification datasets and provide extensive ablation studies for each design choice of our method.

## 2 RELATED WORK

**Fine-Tuning** Fine-tuning is a popular and effective method for adjusting LLMs to specific tasks. Standard fine-tuning (Radford et al., 2019; Brown et al., 2020; Howard & Ruder, 2018; Liu et al., 2019; Lan et al., 2019; Raffel et al., 2020; Sun et al., 2019) retrains the model with new data. However, a key disadvantage is the large number of parameters that must be stored.

**Efficient Fine-Tuning** To alleviate the computational burden of fine-tuning, Adapter-BERT (Houlsby et al., 2019) proposes training only the adapter layers inserted into the model, while BitFit (Zaken et al., 2021) focuses on fine-tuning just the bias terms. Delta Tuning (Ding et al., 2022) explores parameter-efficient methods that adjust only a small portion of a model's parameters. Low-

Rank Adaptation methods (LoRA) (Hu et al., 2021) introduces a novel low-rank adaptation technique, where additional low-rank matrices are added to the weights during training. This allows the model to fine-tune only these matrices, reducing the number of trainable parameters significantly. VERA (Kopiczko et al., 2023) builds on LoRA by incorporating adaptive learning rates. Compacter Karimi Mahabadi et al. (2021) leverages hypercomplex layers, and LoRA-Pro (Wang & Liang, 2024) further refines optimization. Despite these advancements, large models like GPT-3, which contain $175B$ parameters, require updating millions of parameters, such as 17.5M for LoRA.

**Prompt Tuning (PT)** Unlike fine-tuning methods, PT reduces the number of trainable parameters by introducing learnable tokens optimized while keeping the model's weights frozen. Lester et al. (2021) propose appending continuous prompts to the input and optimizing them, while P-tuning (Liu et al., 2023) and Prefix Tuning (Li & Liang, 2021) extend this concept by incorporating learnable tokens at intermediate layers. More recently, Wang et al. (2023) introduced the idea of training a single prompt to be shared across multiple tasks. Although these methods significantly reduce the number of trainable parameters, they face challenges in few-shot learning Gu et al. (2021) and provide limited interpretability for the learned continuous tokens (Ghosal et al., 2024; Khashabi et al., 2021; Deng et al., 2022).

**In-Context Learning (ICL)** In contrast to earlier methods, ICL (Brown et al., 2020) avoids optimization entirely. Instead, it concatenates task-specific examples before the input, allowing the model to learn a new task purely through observation, leveraging its pre-trained knowledge. Despite its advantages, ICL has limitations, often underperforming compared to optimization-based methods (Liu et al., 2022; Peng et al., 2023; Sun et al., 2023).

**Instruction Prompt Tuning (IPT)** IPT (Singhal et al., 2022) combines key elements of PT and ICL, utilizing learnable tokens that are optimized during training alongside static context tokens, similar to ICL. The concept of using both soft and hard prompts was previously introduced by PPT (Gu et al., 2021) and PTR (Han et al., 2022). Yet, IPT has struggled to consistently surpass PT in performance (Sun et al., 2023). While our method shares similarities with IPT, we focus on optimizing context tokens without introducing additional learnable tokens, and we are also leveraging context labels in the process. Another key difference lies in the optimization process, where our loss includes a regularization term, and we employ projected gradient descent to ensure the output stays close to the user-supplied reliable input.

## 3 OUR METHOD

### 3.1 INPUT PREPARATION

Our method takes as input a few-shot classification dataset containing $N$ examples. Each example consists of a pairing of $x$ (an instruction) and $y$ (a label). We embed $(x, y)$ using input, output, and separation templates, converting them into readable text that LLMs better understand, as done in ICL Brown et al. (2020). The input and output templates, denoted $T_i$ and $T_o$, along with separators $S_{\text{intra}}$ and $S_{\text{inter}}$, are provided in appendix E. To embed a single example $(x, y)$ using the template, we concatenate the input $x$ embedded in $T_i$ with $S_{\text{intra}}$, followed by the output $y$ embedded in $T_o$, and finally $S_{\text{inter}}$, resulting in $X_{\text{Emb}_i} = [T_i(x_i), S_{\text{intra}}, T_o(y_i), S_{\text{inter}}]$. To generate the complete context, we concatenate all $X_{\text{Emb}_i}$, forming $X_{\text{Context}} = [X_{\text{Emb}_i}]_{i=1}^{N}$. To construct a complete training example, we randomly select an embedded example from the training set $X_{\text{Emb}_i}$, and concatenate it after the context, resulting $X_{\text{Train}_i} = [X_{\text{Context}}, X_{\text{Emb}_i}]$, which is then fed into the LLM. This process is also visualized in fig. 4. We supply additional concrete examples in appendix G.

Above, we described how we construct a training example $X_{\text{Train}_i}$, as a text sequence. However, before feeding it into the model, we must process the text through a tokenizer, which splits the text into tokens and returns an embedding vector for each token. Each example contains six types of tokens: input, input template, intra-separator, output, output template, and inter-separator. For simplicity, we ignore the separators and the fact that each part usually contains multiple tokens. For each training example $i$ and its sub-example $k$, we focus on four token types: $t_{\text{I}_i}^{(k)}, t_{\text{IT}_i}^{(k)}, t_{\text{O}_i}^{(k)}, t_{\text{OT}_i}^{(k)}$, which represent the input, input template, output, and output template, respectively. Each training example $i$ consists of $N + 1$ sub-examples, $N$ sub-examples in the context and one training sub-example at the end.

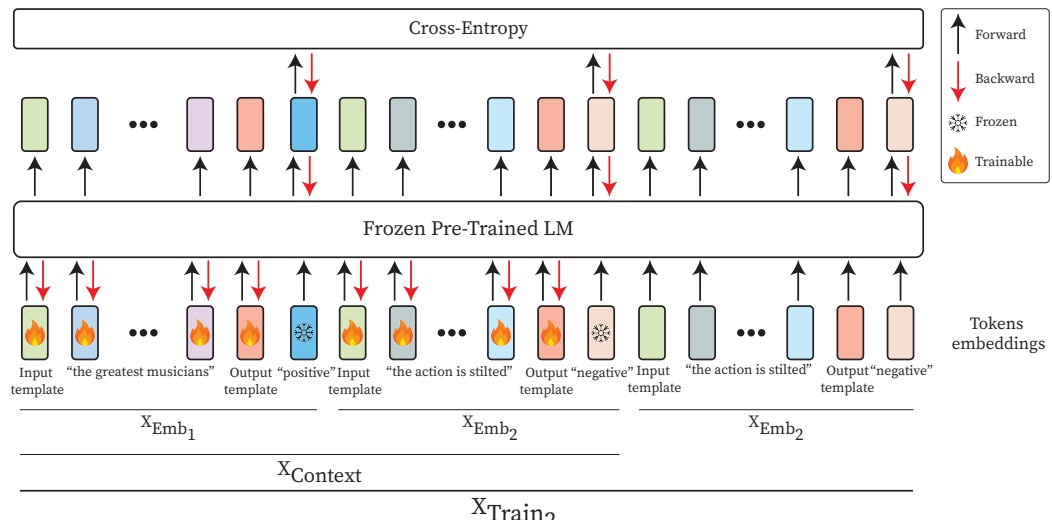

Figure 4: **Overview of CPT Training Process.** We begin by arranging the data. We concatenate all of the training examples that were embedded into the input-output templates $[X_{\text{Emb}_i}]_{i=1}^N$, creating $X_{\text{Context}}$. To this, we append a randomly selected training example, in this case $X_{\text{Emb}_2}$, to form the complete training example $X_{\text{Train}_2}$. For the training process, the input is passed through the frozen LLM, and the loss is calculated using all labels present in $X_{\text{Train}_2}$, covering both the context and training labels. The context is updated, but its labels remain unchanged.

## 3.2 OPTIMIZATION

In this section, we discuss the optimization process of our method, which is inspired by Adversarial Attacks (AT) Madry et al. (2017). Each AT step can be divided into two parts: optimization and restriction. In the first step, an attacker modifies an image to cause incorrect classification, where in the second step, the attack limits its changes to avoid detection. Similarly, our method is split into two parts: optimization, including loss design, as discussed in section 3.2.1, and controlling the token updates, as detailed in section 3.2.2.

### 3.2.1 LOSS DESIGN

The optimization process modifies the input embedding to help the classification. To achieve this, we introduce a new loss for each training example $X_{\text{Train}_i}$. This loss incorporates all the context sub-example labels $X_{\text{Context}}$, or more formally, we use $t_{\text{O}_i}^{(k)}$ for all $k \in [1, N]$. We use these tokens and the model's predictions for those tokens, $\hat{t}_{\text{O}_i}^{(k)}$, as shown in eq. (1).

$$L_{\text{Context}_i} = \sum_{k=1}^{N} \omega_k \cdot \text{CrossEntropy}(\hat{t}_{\text{O}_i}^{(k)}, t_{\text{O}_i}^{(k)}) \tag{1}$$

In addition to $L_{\text{Context}_i}$, we also apply the standard loss on the training sub-example in eq. (2).

$$L_{\text{Train}_i} = \text{CrossEntropy}(\hat{t}_{\text{O}_i}^{(N+1)}, t_{\text{O}_i}^{(N+1)}) \tag{2}$$

Lastly, we sum both losses to create the final loss $L_i = L_{\text{Context}_i} + L_{\text{Train}_i}$, where $L_{\text{Context}_i}$ can be thought of as a regularization for the standard loss $L_{\text{Train}_i}$.

As explained in section 3.1, each training example $X_{\text{Train}_i}$ contains $N + 1$ sub-labels, from $N$ sub-examples in the context and one training sub-example. However, not all sub-examples should be weighted equally. For instance, the last sub-example is more important as it is located in the location of the test examples. Additionally, sub-examples closer to the end of the context carry more

importance (Zhao et al., 2021). Thus, we apply exponential loss weight decay starting from the end of the context and decaying towards the beginning, while keeping $L_{\text{Train}_i}$ unchanged. Formally, each sub-example $k$ is multiplied by $\gamma^j$, where $j = N + 1 - k$. For example, the last sub-example is multiplied by $\gamma^1$, and the second-to-last by $\gamma^2$, and so on. The decay is shown in eq. (1) as $\omega_k$.

### 3.2.2 CONTROLLED TOKEN EMBEDDING OPTIMIZATION

As mentioned in section 3.2.1, we use all the labels in each $X_{\text{Train}_i}$ to optimize the tokens within the context. However, some tokens in the context represent labels, and we do not update these label tokens, as they carry valuable ground truth information. Instead, we update the other tokens in the context, carefully managing these updates to ensure controlled and precise modifications.

The controlled modification is designed with two key objectives. First, we trust the user to provide meaningful examples representing the task, so the context should stay close to the user's intent, minimizing significant changes. Second, few-shot optimization can lead to overfitting without proper regularization. Controlled modification addresses both issues: it acts as a regularization mechanism while preventing overfitting. For instance, as changes become smaller, our method converges to ICL, which is robust against overfitting. We achieve this by using projected gradient descent, which limits each token's embedding change to an $\ell_2$ norm of size $\epsilon$ after each optimization step. Further explanations are provided in appendix H.

## 4 EXPERIMENTAL SETUP

In this section, we provide details regarding the datasets, models, baselines, and evaluation used in our experiments. Implementation details are provided in appendix F.

### 4.1 DATASETS

In this work, we focus on a classification task and select a variety of datasets to ensure robust conclusions across different task types. We include SST-2 (Socher et al., 2013) for sentiment analysis, AG News (Zhang et al., 2015b) for news classification, DBpedia (Zhang et al., 2015a) for ontology classification, and TREC (Li & Roth, 2002) for question classification. These datasets represent a diverse range of natural language classification tasks, include different number of classification classes, allowing us to evaluate our method comprehensively. More details are provided in appendix D.

### 4.2 MODELS

We use models of varying sizes and quality to ensure robust evaluation and conclusions. For the relatively small model, we use BLOOM1.7B (Scao et al., 2022), while for larger models, we opt for GPT-j6B(Wang & Komatsuzaki, 2021) and Llama3 8B(AI@Meta, 2024). The GPT-j model is noted for its robust performance, while Llama3 is currently among the leading models in the field.

### 4.3 BASELINES

We compare our method to several groups of few-shot learning techniques. In the first group, we include LoRA (Hu et al., 2021), one of the leading efficient fine-tuning methods. Additionally, we compare against several prompt-tuning approaches, including Prompt Tuning (PT) (Lester et al., 2021), Prefix Tuning (Li & Liang, 2021), and Instruction Prompt Tuning (IPT) (Singhal et al., 2022). Finally, we compare our method to In-Context Learning (ICL) (Brown et al., 2020).

For some of the few-shot methods, we introduce an alternative version that incorporates instructions, as indicated in table 1 with a †. Instead of initializing the learnable tokens randomly, we initialize them with instructions specified in appendix C. We apply instructions to PT, IPT, and our method, reporting results for both random and instruction-based prompt initialization. An example is provided in appendix G.

Table 1: **Baseline Comparisons** Mean accuracy of various methods and our CPT, across several models and datasets. Evaluations are conducted using 2, 4, and 6 shots.

| Dataset | Method | BLOOM 1.7B | | | GPT-j 6B | | | Llama3 8B | | |
|---|---|---|---|---|---|---|---|---|---|---|
| | | 2 | 4 | 6 | 2 | 4 | 6 | 2 | 4 | 6 |
| SST-2 | Prefix | 47.80 | 47.33 | 49.00 | 52.23 | 52.50 | 52.87 | — | — | — |
| | ICL | 50.53 | 60.83 | 61.87 | 50.57 | 67.47 | 77.47 | 76.43 | 80.63 | 83.10 |
| | PT† | 64.97 | 65.07 | 65.07 | 57.10 | 52.93 | 55.70 | 72.97 | 73.47 | 84.57 |
| | PT | 56.03 | 56.90 | 58.33 | 64.07 | 64.37 | 64.60 | 64.27 | 65.70 | 67.03 |
| | IPT† | 58.50 | 61.83 | 62.80 | 51.50 | **83.20** | 84.80 | 86.90 | 88.03 | 94.40 |
| | IPT | 48.50 | 58.80 | 61.87 | 48.13 | 82.27 | 87.17 | 57.20 | 87.40 | 90.43 |
| | LoRA | **66.40** | 66.93 | 66.90 | **69.80** | 71.53 | 73.17 | 68.73 | 71.27 | 83.97 |
| | CPT† | 59.53 | **72.40** | **74.83** | 52.53 | 82.03 | **88.07** | 92.73 | 95.07 | 96.40 |
| | CPT | 50.77 | 70.70 | 74.10 | 50.53 | 82.90 | 88.03 | 83.83 | **96.30** | **96.50** |
| AG News | Prefix | 24.87 | 25.35 | 26.02 | 32.32 | 33.33 | 46.08 | — | — | — |
| | ICL | 35.12 | 34.28 | 42.48 | 66.73 | 62.38 | 69.57 | 79.38 | 82.32 | 85.27 |
| | PT† | 28.67 | 30.73 | 41.17 | 37.85 | 44.85 | 62.92 | 59.60 | 57.02 | 68.02 |
| | PT | 33.57 | 36.98 | **56.08** | 56.85 | 56.13 | 75.10 | 69.32 | 67.92 | 69.33 |
| | IPT† | 36.95 | 31.90 | 42.93 | 67.02 | 63.00 | 74.85 | 82.93 | **84.45** | 85.08 |
| | IPT | 38.77 | 38.20 | 47.78 | 66.02 | 63.92 | 74.00 | 80.52 | 76.30 | 80.98 |
| | LoRA | 29.50 | 30.80 | 33.98 | 56.12 | 56.03 | 72.55 | 70.62 | 74.97 | 73.70 |
| | CPT† | 33.68 | 33.13 | 41.10 | 71.35 | **68.73** | 75.68 | 83.17 | 84.28 | 84.67 |
| | CPT | **40.85** | **44.48** | 50.40 | **74.80** | 68.62 | **76.22** | **83.78** | 81.92 | **85.43** |
| DBpedia | Prefix | 19.76 | 19.74 | 23.65 | 13.25 | 16.43 | 24.94 | — | — | — |
| | ICL | 48.20 | 51.40 | 55.17 | 50.87 | 62.46 | 70.76 | 71.66 | 72.44 | 79.93 |
| | PT† | 24.90 | 26.32 | 34.75 | 21.01 | 22.12 | 37.44 | 55.30 | 57.21 | 66.26 |
| | PT | 46.71 | 41.94 | 45.93 | 23.39 | 29.69 | 40.53 | 55.81 | 52.72 | 55.02 |
| | IPT† | 33.28 | 40.36 | 45.85 | 47.10 | 67.60 | 75.09 | 81.10 | 87.69 | 92.06 |
| | IPT | 48.09 | 54.60 | 70.57 | 52.86 | 67.27 | 70.73 | 72.92 | 76.11 | 78.44 |
| | LoRA | 43.30 | 41.13 | 41.18 | 30.15 | 28.02 | 41.50 | 54.24 | 59.50 | 63.21 |
| | CPT† | 33.80 | 48.13 | 51.18 | 53.20 | **77.30** | **81.00** | **84.23** | **90.33** | **93.08** |
| | CPT | **58.85** | **65.78** | **73.55** | **68.29** | 75.07 | 77.65 | 77.38 | 78.49 | 82.42 |
| TREC | Prefix | 19.10 | 24.49 | 29.92 | 30.76 | 30.04 | 27.87 | — | — | — |
| | ICL | 33.54 | 33.33 | 28.53 | 28.94 | 35.14 | 32.49 | 35.32 | 42.48 | 40.34 |
| | PT† | 30.91 | 33.70 | 39.31 | 29.02 | 34.66 | 43.89 | 43.42 | 48.81 | 51.73 |
| | PT | 32.18 | 32.26 | 35.69 | 31.16 | 32.79 | 37.86 | 32.77 | 33.98 | 33.83 |
| | IPT† | 27.83 | 36.64 | 42.92 | 31.04 | 43.12 | 43.09 | 51.72 | 62.14 | 65.13 |
| | IPT | 32.37 | 36.59 | 42.60 | 29.59 | 38.90 | 40.38 | 36.94 | 45.62 | 52.08 |
| | LoRA | 34.07 | 33.22 | 33.50 | 34.17 | 33.73 | 37.63 | 31.21 | 33.21 | 36.36 |
| | CPT† | 29.72 | 35.64 | **45.38** | 33.39 | 44.20 | **45.83** | 57.26 | **67.00** | **69.29** |
| | CPT | **35.68** | **41.79** | 45.16 | **35.37** | **44.66** | 42.71 | 45.12 | 57.54 | 60.18 |

## 4.4 EVALUATION

We evaluate each model and dataset using three different numbers of training samples: 2, 4, and 6. For each configuration, the reported results are averaged accuracy over 30 experiments, consisting of 10 randomly sampled templates and 3 different random seeds, with the templates described in appendix E. By utilizing randomized seeds, we ensure variation in the selection of training examples. This extensive setup is crucial for achieving a comprehensive and robust evaluation, especially given that these methods are known to be highly sensitive to the selection of training examples and templates (Voronov et al., 2024; Zhao et al., 2021). Further evaluation details can be found in appendix B.

## 5 RESULTS

### 5.1 MAIN RESULTS

In table 1, we demonstrate that CPT convincingly performs better than the baselines in most cases, with particularly pronounced gains in harder tasks. Furthermore, CPT 's performance becomes more efficient and effective as the models grow stronger, such as with Llama3.

**Performance on Challenging Tasks** CPT demonstrates improvements across various datasets, with more pronounced gains in tasks we define as harder based on two factors: the number of shots and the number of classes. As illustrated in table 1, task difficulty increases with fewer shots and more classes. For example, on the DBpedia dataset, which has 14 classes, decreasing the shots from 6 to 4 widens the performance gap between CPT and the baselines from (3, 6, 1) to (11, 10, 3) across the models: BLOOM, GPT-j, and Llama3.

**Decisive Advantage with Powerful Models** The strength of the model plays a significant role in performance. As the model becomes better, CPT's advantage becomes more pronounced across all datasets and shot settings. For instance, Llama3 consistently outperforms other baselines across all datasets, except in one case where results are comparable. With GPT-j, a slightly older model, the results are lower in two instances, with one comparable outcome, both on SST-2 , the easier task as previously discussed. When comparing with BLOOM , the weakest model in our comparison, we observe lower performance on two occasions, specifically on the two easier datasets.

### 5.2 STANDARD DEVIATION

Standard deviation (std) plays a crucial role in few-shot learning due to the sensitivity of these methods to both the training examples and the chosen template (Zhao et al., 2021; Voronov et al., 2024). In fig. 5, we present accuracy along with two types of std bars: black bars represent the mean std across different templates, while blue bars represent the mean std across different seeds. We demonstrate that CPT significantly improves accuracy across various models and datasets in a statistically significant manner. More information is presented in appendix A.

Our method's standard deviation performs equivalently to other methods in most cases, while in certain cases, such as with DBpedia, CPT exhibits both higher accuracy and lower std, reinforcing its robustness in complex tasks. However, the sensitivity of our method does not follow a clear pattern across random seeds or templates. For instance, while randomness in templates and training examples has an equal influence on std in DBpedia and TREC , SST-2 shows a higher std for template randomness, and AG News is more sensitive to variations in training examples.

### 5.3 ABLATIONS

The most important design choices that positively impacted CPT's performance are the loss design and the projections. These improvements are evident across 2, 4, and 6-shot settings, as shown in

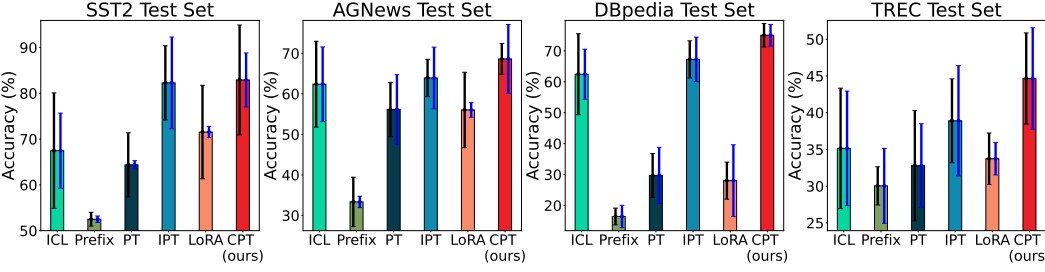

Figure 5: **Accuracy and Standard Deviation** Comparison of accuracy and standard deviation between CPT and baselines, evaluated with 4-shot on GPT-j model. The black bars represent the mean std across different templates, while the blue bars represent the mean std across different seeds.

Table 2: **Ablation Study** We present the mean accuracy for various ablations using the GPT-j model and the DBpedia dataset, including loss tokens (train example, random, or all context), loss weighting (decay and mean), projection type (token-wise or all-tokens), epsilon values for input and format, updated tokens (input, format, masks), and masking of the training example.

| Loss Tokens | Loss Weighting | Projection Type | Input $\epsilon$ | Format $\epsilon$ | Updated Tokens | Mask Training Example | Number of Training Examples | | |
|---|---|---|---|---|---|---|---|---|---|
| | | | | | | | 2 | 4 | 6 |
| Train Example | | | | | | | 58.09 | 61.54 | 66.69 |
| Train Example & 1 Random | Decay 0.95 | Token-Wise | 0.1 | 0.1 | Input & Format | ✗ | 69.48 | 72.08 | 76.80 |
| Train Example & All Context | | | | | | | 69.54 | 73.03 | 76.58 |
| | Mean | | | | | | 69.62 | 72.91 | 76.49 |
| | Equal 1 | | | | | | 69.07 | 72.82 | 76.23 |
| Train Example & All Context | Equal 10 | Token-Wise | 0.1 | 0.1 | Input & Format | ✗ | 69.35 | 71.01 | 75.11 |
| | Decay 0.99 | | | | | | 69.59 | 72.97 | 76.43 |
| | Decay 0.95 | | | | | | 69.54 | 73.03 | 76.58 |
| | Decay 0.5 | | | | | | 69.60 | 72.39 | 76.44 |
| | | | 0.001 | - | | | 51.52 | 63.41 | 71.50 |
| Train Example & All Context | Decay 0.95 | All-Tokens | 0.01 | - | Input & Format | ✗ | 56.37 | 68.12 | 73.66 |
| | | | 0.1 | - | | | 69.51 | 72.64 | 76.06 |
| | | | 1.0 | - | | | 63.11 | 64.78 | 71.94 |
| | | | 0.01 | 0.1 | | | 65.61 | 70.12 | 75.63 |
| | | | 0.1 | 0.1 | | | 69.54 | 73.03 | 76.58 |
| Train Example & All Context | Decay 0.95 | Token-Wise | 1.0 | 0.1 | Input & Format | ✗ | 65.29 | 66.30 | 73.63 |
| | | | 0.1 | 0.01 | | | 69.53 | 73.55 | 76.55 |
| | | | 0.1 | 1.0 | | | 68.27 | 71.91 | 68.27 |
| | | | | | Input | | 69.47 | 74.13 | 76.63 |
| Train Example & All Context | Decay 0.95 | Token-Wise | 0.1 | 0.1 | Masks | ✗ | 63.74 | 69.21 | 74.91 |
| | | | | | Input & Format | | 69.54 | 73.03 | 76.58 |
| Train Example & All Context | Decay 0.95 | Token-Wise | 0.1 | 0.1 | Input & Format | ✓ | 67.55 | 64.26 | 68.58 |

table 2. Different options for the loss design are specified under "Loss Tokens", with three configurations: using only the training label, using the training label plus one random context label, and using the training label plus all context labels. The latter significantly outperforms the training-only configuration. The ablation over the projection is specified under "Input $\epsilon$" and "Format $\epsilon$" demonstrating that both too small a change (which converges to ICL) and too large a norm are suboptimal. Lastly, we examined the effect of "Loss Weighting". We propose three options: Mean, which applies uniform weighting across all labels; Equal, which assigns equal weight to both the training label loss and the context label losses; and the option we use, Decay, which reduces the influence of context labels further from the training example. On this dataset, Decay works slightly better, and in most cases, the improvement is more significant.

In addition to these core design choices, we explored several alternative configurations that ultimately did not enhance performance. Under "Loss Weighting", we experimented with the "Equal" option, which assigns equal loss weight to both the training example loss and the entire context loss, where the training loss can be multiplied by the noted value (*e.g.*, 1, 10). We also tested the projection type "All-Tokens" which applies the projection to the entire context collectively rather than token-by-token. Under "Updated Tokens" we attempted to modify only specific parts of the context. Additionally, under "Mask Training" we masked the training example from the context to prevent the model from simply copying the answer. However, none of these configurations led to performance improvements. Additional ablation experiments are presented in appendix I.

## 6 DISCUSSIONS

CPT demonstrates significant advancements in few-shot learning by integrating ICL with PT and adversarial strategies, refining context embeddings. Unlike traditional fine-tuning and other parameter-efficient approaches, CPT optimizes only the context tokens, making it particularly effective in few-shot settings, where overfitting is a concern. CPT achieves improved generalization across a wide variety of tasks, demonstrates a significant advancement, and offers meaningful insights for future few-shot learning methods.

**Limitation & Future Work** The computational cost associated with the iterative optimization of context embeddings is significant compared to ICL. Additionally, similar to ICL and IPT, CPT is limited in the number of examples it can handle, as memory consumption scales with context length. In contrast, traditional methods are better suited for larger datasets. Future work could explore more efficient optimization strategies to reduce computational overhead and improve scalability.

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
