# OpenReview forum: "Context-aware Prompt Tuning: Advancing In-Context Learning with Adversarial Methods"
_ICLR.cc/2025/Conference — Submitted to ICLR 2025_

### Official Review · Reviewer_iktw · 2024-11-01

**Soundness:** 2
**Presentation:** 2
**Contribution:** 2
**Rating:** 5
**Confidence:** 4

**Summary:**

This paper proposes context-aware prompt tuning (CPT), a few-shot learning method that optimizes the in-context example using projected gradient descent (PGD). The authors conduct experiments on various models and benchmarks, finding CPT outperforms other baseline methods (prompt tuning, LoRA, instruction prompt tuning).

**Strengths:**

The idea of optimizing the context using PGD is novel. If one needs to optimize the in-context example, it makes sense to use PGD so that the examples will not change by too much and the optimization does not overfit the few-shot examples.

**Weaknesses:**

1. It is unclear what is the motivation for optimizing the in-context example. Does that make more sense than optimizing the prompt in instruction prompt tuning (IPT)? There is one possibility that the performance should be attributed to the use of PGD, which might prevent from overfitting. It will be beneficial if there is abation study on that, e.g., uses other regularized optimizers or use PGD in baselines (IPT, PT).

2. Lack of analysis on the experimental results. Again, it might make the results easier to analysize if the authors disentangle these two components: (i) train on in-context samples; (ii) PGD as the optimizer.

3. Some paragraphs are not clearly written. For example, in section 4.3, the authors introduce another variant $\dagger$ which initializes trainable tokens with human engineered instructions. It is unclear what is the difference between $PT^\dagger$ and $IPT^\dagger$ for few-shot setting since they both have few-shot examples and trainable instruction-initialized tokens.

4. The Set Classification dataset is confusing. I feel the only mechanism that works for an input sequence from that dataset is to search in the context to see if the test example appears and copy the cooresponding label (if any) as the output (please correct me if I am wrong), which is simply an induction head,  and the loss design of CPT seems to be encouraging the forming of induction head. Therefore is it a good benchmark to measure few-shot learning?

5. In section 5.1 the title "Better with Harder Tasks" for that paragraph seems to be overclaiming.

**Questions:**

Please see weakness.

---

> ### Author Response · Authors · 2024-11-20
> **answer to iktw [1/3]**
>
> We appreciate the reviewer’s recognition of the novelty of our approach, particularly the use of projected gradient descent (PGD) for optimizing the context in few-shot learning scenarios. As noted, PGD offers a principled way to optimize in-context examples while ensuring that changes remain minimal. This not only preserves the original structure of the examples but also prevents the optimization process from overfitting to the limited few-shot examples, addressing one of the key challenges in few-shot learning.
>
> In addition, we would like to highlight several other contributions of our work. We propose a simple yet effective method that integrates key elements from existing approaches in a well-motivated manner. Our approach is built on the observation that existing methods, such as in-context learning (ICL) and fine-tuning, are not optimal for few-shot learning tasks. ICL is highly sensitive to the choice and ordering of examples, while fine-tuning often suffers from overfitting — a challenge that, to the best of our knowledge, is identified here for the first time in this context.
>
> Our method combines the strengths of both approaches. Like ICL and unlike Prompt Tuning (PT), we initialize with examples; like PT and unlike ICL, we optimize these tokens. While optimizing the context tokens, we keep the associated labels fixed, as they carry meaningful information. Additionally, we design a novel loss function that incorporates the standard loss (as in PT) while introducing a new term to account for the context information. This loss weights each token using an exponential decay, leveraging the "recency bias" phenomenon.
>
>
> To ensure meaningful updates, we employ projected gradient descent, a technique commonly used in adversarial attacks, to keep the updated tokens close to their initialization under the assumption that the user-provided input is inherently valuable. Finally, to validate the robustness of our method, we introduce a new dataset excluded from the LLM’s training data. This dataset further demonstrates the effectiveness of our approach in improving performance across a variety of tasks.
> We also conducted an extensive ablation study to analyze the contributions of each component of our method, which we will elaborate on later. This study provides valuable insights into the effectiveness and interplay of the design choices in our approach.
>
>
> “unclear what is the motivation for optimizing the in-context example”
> The motivation for optimizing the in-context examples is to address the overfitting phenomenon. While PT updates all prefix tokens, our method initializes the prefix as a context, similar to ICL. Recognizing the overfitting problem illustrated in Fig. 1, we were motivated to modify the loss term and incorporate a novel regularization. Since our context includes both the examples and their corresponding ground truth labels, we aimed to leverage this additional, verified information within the context.
> To put it differently, rather than focusing solely on being correct for the train example (as with the standard loss of PT), we aim to update the tokens in a way that improves predictions for all examples within the context along with the train example. This broader approach ensures that the optimization captures the structure of the task more effectively.
>
>
> We further validated this hypothesis through an ablation study (Tab. 2, first chunk). In this study, we evaluated different loss terms: including only the standard loss (used in PT), including the standard loss and the loss of one random example from the context, and including the standard loss along with all examples in the context(as used by our method). The results clearly show that including one random example improves performance, and incorporating all context examples significantly enhances it, confirming our intuition.
>
> "Does that make more sense than optimizing the prompt in instruction prompt tuning (IPT)?”
> Our solution can be observed from different angles. It can be viewed as an extension of In-Context Learning (ICL), enhanced by integrating elements of Prompt Tuning (PT) and Projected Gradient Descent (PGD), which is the perspective we emphasize in this work. Alternatively, it can be seen as starting with Instruction Prompt Tuning (IPT), but with a modified loss function and the ability to update all tokens. Unlike IPT, which updates only the learnable tokens, our approach optimizes the context itself. (We hope we understood your question correctly!)

---

> ### Author Response · Authors · 2024-11-20
> **answer to iktw [2/3]**
>
> We answer two comments:
>
> “There is one possibility that the performance should be attributed to the use of PGD, which might prevent from overfitting. It will be beneficial if there is abation study on that.”
>
> and
>
> “Lack of analysis on the experimental results. Again, it might make the results easier to analysize if the authors disentangle these two components: (i) train on in-context samples; (ii) PGD as the optimizer.”
>
>
> Thank you for raising this concern—it is indeed an important experiment, and we have included it in the updated version of the paper (App. I). However, it is important to emphasize once again that the improvements achieved by our method do not solely depend on PGD, as previously discussed. In the our answer for “unclear what is the motivation for optimizing the in-context example,” we demonstrated that our novel loss function also significantly contributes to the enhancement of performance.
> It is important to mention that we used the same optimizer for all baselines, including our method. However, our method incorporates an additional step after each parameter update: we project each token, restricting its allowed change. The allowed change is determined by the hyperparameters Input \epsilon and Format \epsilon, which define the L2 norm limit for each token’s modification.
>
>
> To ensure that PGD is not the sole reason for our improvement, we conducted two types of experiments, as you suggested. First, we compared our method without PGD to PT and IPT. Second we added a PGD step to PT and IPT for comparison.
> For the first experiment, we compared CPT (without PGD) to PT and IPT on DBpedia. The results for 2, 4, and 6 shots are as follows:
>
>
> PT 23.39, 29.69, 40.53
>
> IPT 52.86, 67.27, 70.73
>
> CPT 68.28, 74.17, 77.52
>
>
> For the second experiment, we compared CPT\dag to PT\dag and IPT\dag (with and without PGD), using DBpedia. To ensure a fair comparison, we performed hyperparameter tuning (HPT) over \epsilon and over the learning rate for both PT and IPT. The results for 2, 4, and 6 shots are as follows:
>
>
> PT\dag             12.96, 22.12, 37.44  (LR 1e-3)
>
> PT\dag +PGD  13.08, 22.02, 38.69     (LR 1e-3, epsilon=1.0)
>
> IPT\dag            47.10, 66.37, 75.09   (LR 1e-5)
>
> IPT\dag +PGD 47.10, 66.40, 75.09      (LR 1e-5, epsilon=0.1)
>
> CPT\dag           52.87, 77.30, 81.00   (LR 1e-5, input epsilon=0.1, format epsilon=0.1)
>
>
>
> The results clearly demonstrate that, in both experiments, our method consistently outperforms PT and IPT. Furthermore, it is evident that other methods do not necessarily benefit from the addition of PGD. While we cannot definitively explain this, we hypothesize that it may be due to the highly effective way in which we employ PGD, leveraging prior knowledge about the structure of the input, format, and labels within the context. Our approach allows us to apply distinct projections to different components of the context, which we believe significantly contributes to the superior performance of our method.
>
>
>
> “It is unclear what is the difference between PT \dag and IPT \dag.”
> Apologies for the confusion. To clarify this issue, we have added an example in the updated version of the paper (App. G). In Prompt Tuning (PT) and Instruction Prompt Tuning (IPT), there are learnable tokens that, when using the \dag variant, are initialized with the instruction. However, unlike PT, IPT also includes the context, which follows the instruction and remains fixed, without being updated during training.
> For example, for SST2 the instruction is “Classify the sentiment of the following text as positive or negative”. And lets take two instances from the dataset: Example 1: “input: the greatest musicians output: positive” Example 2: “input: the action is stilted: negative”.
>
> For PT the prefix is: “Classify the sentiment of the following text as positive or negative” which is updated during training.
>
> For IPT the prefix is “Classify the sentiment of the following text as positive or negative. input: the greatest musicians output: positive. input: the action is stilted: negative” and only the instruction are updated during training, only this part “Classify the sentiment of the following text as positive or negative”.

---

> > ### Comment · Reviewer_iktw · 2024-11-21
> >
> > I appreciate the ablation study. Just a quick question, what model are you using in the ablation study?

---

> ### Author Response · Authors · 2024-11-20
> **answer to itkw [3/3]**
>
> “Set Classification dataset is confusing”
> While we acknowledge that the Set Classification dataset might not be the ideal benchmark, it is only one of the five datasets used to validate the effectiveness of our method. Its inclusion adds diversity to the evaluation and highlights different aspects of the method's performance. The updated explanation is included in the paper under the "Set Classification" section, with additional information provided in Appendix A.
>
>
> Regarding your concern that the task might encourage searching in the context and copying the corresponding label, we agree this might be a factor. However, we believe that plain copying is insufficient to excel in this task due to the way the dataset is constructed. As detailed in Appendix 1, the dataset’s design ensures that classification cannot be trivially reduced to such a mechanism. For instance, in the creation of Fig. 5, we use 10 groups, each containing 5 members, and each example includes 4 words from a single group.
>
> In the 2-shot setting, only 2 groups are seen, and even for these groups, not all 5 words are observed. Additionally, the order of the words in the input can vary while the labels remain consistent, reducing reliance on a strict inductive head property. This setup ensures the task remains challenging in low-shot scenarios, requiring additional behaviors to be learned to answer the task properly rather than relying on plain copying.
>
>
>
> In section 5.1 the title "Better with Harder Tasks" for that paragraph seems to be overclaiming.
> Thank you for pointing this out. We understand that the title "Better with Harder Tasks" might come across as overclaiming. To better reflect the content of the section, we have revised the title to "Performance on Challenging Tasks", which more accurately describes the discussion. This revised title emphasizes how our method performs effectively on more challenging tasks without implying absolute superiority. We appreciate your feedback, as it helps ensure the clarity and accuracy of our claims.

---

> > ### Comment · Reviewer_iktw · 2024-11-21
> > **Set Classification dataset still does not make sense to me**
> >
> > Thanks for the response. The Set Classification dataset still does not make sense to me and I feel there is some misunderstanding.
> >
> > I actually agree that it is a challenging task only using an induction head if the number of in-context examples is small. It's challenging because for the few-shot learning, most likely the query in the sequence is never seen in the context.
> > The issue is that I don't think it's chanllenging in a reasonable way. How should a model make any "reasonable" prediction to the query example if it is never seen in the context? The words and their labels are randomly grouped/assigned and there are no semantic relations between them. In this case, the only prediction that is reasonable is those from an induction head -- basically just copy and paste from the context if it was seen in the context. Otherwise, there should not be any reasonable way to make a prediction for the query.
> >
> > In this sense, I don't feel the Set Classification dataset is a good benchmark for few-shot learning.

---

> > > ### Author Response · Authors · 2024-11-22
> > > **Replying to iktw**
> > >
> > > We believe that addressing your concerns has clarified and improved the quality of the paper, and we thank you for your valuable feedback that contributed to these enhancements. In light of this, we kindly ask you to consider revising your rating to reflect these improvements.

---

> ### Author Response · Authors · 2024-11-22
> **Replying to iktw - Ablation**
>
> Ablation
>
> Apologies for missing this detail earlier. We used GPT-J for all the ablations, including those in question.

---

> ### Author Response · Authors · 2024-11-22
> **Replying to iktw - Set Classification Dataset**
>
> Thank you for your continued engagement and detailed feedback. After carefully reflecting on your concerns regarding the Set Classification dataset, we recognize that its lack of semantic structure and reliance on random groupings might make it less representative of real-world few-shot learning tasks. While the dataset was initially included to highlight the adaptability of CPT to unconventional tasks, we agree that its relevance as a benchmark may be limited.
>
> Given this, we are open to removing the dataset from the paper and focusing our evaluation on the remaining four benchmarks (AGNews, SST-2, DBpedia, and TREC). These datasets represent more traditional few-shot learning scenarios and robustly demonstrate the strengths of CPT across diverse tasks.

---

> > ### Comment · Reviewer_iktw · 2024-11-22
> >
> > Given the response, I increased the score.

---

### Official Review · Reviewer_DESR · 2024-11-03

**Soundness:** 3
**Presentation:** 3
**Contribution:** 3
**Rating:** 6
**Confidence:** 4

**Summary:**

This paper proposes a few-shot method called context-aware prompt tuning. This method is inspired by in-context learning and prompt tuning, concatenating examples before the input as in ICL and learning these examples as in prompt tuning. The authors verify the effectiveness of the proposed context-aware prompt tuning method on several classification tasks using three models.

**Strengths:**

The proposed method is straightforward and easy to understand.
The experiments in this paper are comprehensive, which use three models and test on several classification datasets.
The results show that the proposed method outperforms LoRA, PT, IPT and ICL in most cases.

**Weaknesses:**

1. The proposed method integrates ICL and PT intuitively, thus the novelty is not very significant.
2. Since the proposed context-aware prompt tuning is an optimization-based method, it also has the overfitting problem, just like fine-tuning and PT. Thought authors tried to mitigate overfitting by incorporating context labels into the loss function and applying projected gradient descent, the overfitting problem still exists in context-aware prompt tuning.

**Questions:**

1. Why ICL cannot fully extract the information that exists in the training examples?
2. In Table 1, CPT† (incorporating instructions) outperforms CPT on some datasets/models (for example SST-2 with BLOOM 1.7B), but underperforms CPT on other settings (for example DBpedia with BLOOM 1.7B, decreasing from 58.85 to 33.80 using 2 shots). How to explain this phenomenon?

---

> ### Author Response · Authors · 2024-11-18
> **answer to DESR [1/2]**
>
> Thank you for appreciating the clarity of our proposed method, noting that it is easy to understand. We are also grateful for your acknowledgment of our comprehensive experiments, which employed three models and tested on multiple classification datasets. Additionally, we are pleased that you highlighted the results demonstrating our method's superiority, as it outperforms LoRA, PT, IPT, and ICL in most cases.
>
>
> "The proposed method integrates ICL and PT intuitively, thus the novelty is not very significant"
> We propose a simple yet effective method that integrates key elements from existing approaches in a well-motivated manner. We present a solid motivation by demonstrating that current methods, such as in-context learning (ICL) and fine-tuning, are not optimal for few-shot learning tasks. ICL is highly sensitive to the choice of the examples, while fine-tuning suffers from overfitting — an issue that, to the best of our knowledge, is raised here for the first time in this context.
>
> Our method combines the strengths of both approaches. Like ICL and unlike Prompt Tuning (PT), we initialize with examples. Like PT and unlike ICL, we optimize these tokens. While optimizing the context tokens, we keep the associated labels fixed, as they carry meaningful information. In addition, we design a novel loss function that includes the standard loss (as in PT) but also introduces a new term that accounts for the context information. This loss weights each token using an exponential decay, leveraging the "recency bias" phenomenon.
>
> To ensure meaningful updates, we use projected gradient descent, a technique commonly used in adversarial attacks, to keep the updated tokens close to their initialization, assuming the user-provided input is inherently valuable. Finally, we introduce an additional dataset, excluded from the LLM's training data, to further validate our method's improvements and ensure its robustness.
>
>
> "the overfitting problem still exists in context-aware prompt tuning"
> We acknowledge that the overfitting problem still exists, and we agree with your observation. We did not claim otherwise in the paper. Any optimization process guides the model toward the minima of the loss function, but there is no guarantee of reaching a global minimum. Regularization techniques are commonly used to help optimization avoid poor local minima, and we adopted similar strategies by leveraging prior knowledge about the task.
>
> In the paper, we identified overfitting as a significant challenge in few-shot learning, as illustrated in Fig. 1, and hypothesized that mitigating overfitting would enhance performance. To address this, we proposed Context-Aware Prompt Tuning (CPT), which introduces additional loss terms and employs PGD. Our hypothesis was validated through experiments conducted across multiple datasets and models, demonstrating that CPT effectively addresses this challenge.

---

> ### Author Response · Authors · 2024-11-18
> **answer to DESR [2/2]**
>
> "Why ICL cannot fully extract the information that exists in the training examples"
> We agree that the sentence was not well written, and we have revised it in the abstract from “While ICL is not prone to overfitting, it does not fully extract the information that exists in the training examples” to: “ In contrast, ICL avoids overfitting but typically underperforms compared to optimization-based methods and is highly sensitive to the selection, order, and format of demonstration examples.” What we originally meant is that the performance of ICL is generally inferior to optimization-based methods, a claim supported by [1,2,3].
>
> [1] Haokun Liu et al., Few-shot parameter-efficient fine-tuning is better and cheaper than in-context learning.
>
> [2] Hao Peng et al., When does in-context learning fall short and why? A study on specification-heavy tasks.
>
> [3] Simeng Sun et al., How does in-context learning help prompt tuning?
>
>
>
> "How to explain this phenomenon"
> This is indeed an interesting result that we have also explored, but unfortunately, we currently only have hypotheses without definitive proof. Nonetheless, we are happy to share our thoughts on the matter.
> First, we considered the possibility of a bug in our implementation, as we initially expected instructions to consistently improve performance across all datasets. However, this trend is not unique to our implementation—it is also observed in both Prompt Tuning (PT) and Instruction Prompt Tuning (IPT), particularly when comparing Bloom’s performance on SST-2 and DBpedia. This suggests that the observed behavior is more likely attributable to inherent model and dataset characteristics.
>
> Another consideration is the robustness and suitability of the instructions themselves. While instructions improve Bloom's performance on SST-2, they do not have the same effect on DBpedia. However, these same instructions significantly enhance LLaMA3's performance on DBpedia, highlighting potential differences in how models utilize instruction-based information.
> Our most probable hypothesis is based on a well-known property of LLMs: their ability to handle input sequence lengths varies across models. For SST-2, the task involves only two classes, and each example has a shorter input length, resulting in shorter prefixes. In contrast, DBpedia includes 14 classes and longer example input lengths (up to 6 times longer than SST-2), leading to significantly longer prefixes. Models like LLaMA3, which are better equipped to handle longer sequences than Bloom, are expected to perform better on DBpedia. Conversely, Bloom seems to benefit from shorter sequences, which explains its improved performance on SST-2 when instructions are included.

---

### Official Review · Reviewer_CoLs · 2024-11-04

**Soundness:** 2
**Presentation:** 2
**Contribution:** 2
**Rating:** 5
**Confidence:** 4

**Summary:**

This paper introduces an approach to mitigating overfitting in few-shot learning scenarios, where traditional fine-tuning often leads to overfitting, and in-context learning (ICL) performance is highly sensitive to the choice of demonstration examples. The authors propose a context-aware prompt tuning strategy that integrates elements of ICL and prompt tuning. In this method, the demonstration examples provided to the model are treated as tunable parameters, while the associated labels remain fixed. By tuning the context of these demonstrations rather than the entire model, the approach leads to improved generalization across multiple benchmarks.

**Strengths:**

- Proposes a simple extension of prompt tuning by combining it with in-context learning.
- Evaluates the model on different open-source datasets like AGNews, SST-2, DBpedia and TREC.
- The figures (fig 2) in the paper are really helpful in understanding key differences between their method and various baselines they implemented.

**Weaknesses:**

- The idea introduced is very similar to prompt tuning and the whole paper is just about applying it in-conjunction with in-context learning making it harder to find novelty in the approach.
- The writing of the paper can be improved by quite a bit eg: abstract of this paper is hard to understand talking about different approaches rather than making it more crux on the solution/method and same applies to description provided on the set classification dataset.

**Questions:**

- Was any hyperparameter tuning or prompt optimization conducted for the baseline models? The reported performance of Llama-3 seems surprisingly low on straightforward tasks like SST-2, where zero-shot models typically perform well. This raises questions about whether the baseline results fully reflect the model's potential or if they were hindered by suboptimal configurations. Clarification on any tuning efforts for the baselines would be helpful to ensure a fair comparison with the proposed method.

---

> ### Author Response · Authors · 2024-11-17
> **answer to CoLs [1/2]**
>
> We sincerely appreciate your thoughtful review and recognition of the contributions of our work. You highlighted the simplicity of our approach, describing it as an effective extension of prompt tuning that integrates in-context learning (ICL). This balance between simplicity and innovation is a key strength of our method. Additionally, we are pleased that you found our experimental evaluation thorough, as we tested our method on diverse open-source datasets, including AGNews, SST-2, DBpedia, and TREC, demonstrating its robustness across a variety of classification tasks. Finally, we are glad that you found Fig. 2 particularly helpful in illustrating the key distinctions between our method and the baseline approaches, as providing clear and accessible visualizations was a priority in presenting our work.
>
> We now proceed to address each of your concerns in detail, ensuring that all your points are thoroughly clarified and resolved.
>
>
>
> “making it harder to find novelty in the approach”
> We propose a simple yet effective method that selects and integrates key elements from existing approaches in a well-motivated manner. We present a solid motivation by demonstrating that current methods, such as in-context learning (ICL) and fine-tuning, are not optimal for few-shot learning tasks. ICL is highly sensitive to the choice of demonstration examples, while fine-tuning suffers from overfitting — an issue that, to the best of our knowledge, is raised here for the first time in this context.
>
> Our method combines the strengths of both approaches. Like ICL and unlike Prompt Tuning (PT), we initialize with examples. Like PT and unlike ICL, we optimize these tokens. While optimizing the context tokens, we keep the associated labels fixed, as they carry meaningful information. In addition, we design a novel loss function that includes the standard loss (as in PT) but also introduces a new term that accounts for the context information. This loss weights each token using an exponential decay, leveraging the "recency bias" phenomenon.
>
> To ensure meaningful updates, we use projected gradient descent, a technique commonly used in adversarial attacks, to keep the updated tokens close to their initialization, assuming the user-provided input is inherently valuable. Finally, we introduce an additional dataset, excluded from the LLM's training data, to further validate our method's improvements and ensure its robustness.
>
> "abstract of this paper is hard to understand talking about different approaches rather than making it more crux on the solution/method"We have revised this part and updated the paper accordingly. Below is the updated version:
>
> Large Language Models (LLMs) can perform few-shot learning using either optimization-based approaches or In-Context Learning (ICL). Optimization-based methods often suffer from overfitting, as they require updating a large number of parameters with limited data. In contrast, ICL avoids overfitting but typically underperforms compared to optimization-based methods and is highly sensitive to the selection, order, and format of demonstration examples. To overcome these challenges, we introduce Context-aware Prompt Tuning (CPT), a method inspired by ICL, Prompt Tuning (PT), and adversarial attacks. CPT builds on the ICL strategy of concatenating examples before the input, extending it by incorporating PT-like learning to refine the context embedding through iterative optimization, extracting deeper insights from the training examples. Our approach carefully modifies specific context tokens, considering the unique structure of the examples within the context.In addition to updating the context with PT-like optimization, CPT draws inspiration from adversarial attacks, adjusting the input based on the labels present in the context while preserving the inherent value of the user-provided data. To ensure robustness and stability during optimization, we employ a projected gradient descent algorithm, constraining token embeddings to remain close to their original values and safeguarding the quality of the context. Our method has demonstrated superior accuracy across multiple classification tasks using various LLM models, outperforming existing baselines and effectively addressing the overfitting challenge in few-shot learning.

---

> ### Author Response · Authors · 2024-11-17
> **answer to CoLs [2/2]**
>
> “same applies to description provided on the set classification dataset” As requested by R4, we have removed the Set Classification dataset from the paper
>
>
> “Clarification on any tuning efforts for the baselines would be helpful to ensure a fair comparison with the proposed method”
> Yes, we did perform hyperparameter tuning (HPT) for the baselines. Conducting full HPT for all baselines was computationally challenging due to the large parameter space. However, ensuring a fair comparison was always a priority in our experiments. Specifically, we used the default parameters for each method and performed HPT for the learning rate of each baseline, as detailed in Appendix F1. The implementation details for each baseline are presented in Appendix F2, and the training specifics are provided in Appendix F3.
>
> "The reported performance of Llama-3 seems surprisingly low on straightforward tasks like SST-2"
> We stand behind these results and, and the validity and faithfulness of our results have been a central focus throughout our work. We ensured that the reported results fall within the same range as those reported in prior studies, such as Voronov et al. [1] in Appendix F, Table 10. In [1], the authors reported results using LLaMA3-Instruct with ICL on 2-shot settings, while our work uses LLaMA3 with ICL on 2-shot settings. The comparison is as follows:
>
> Ours (LLaMA3, ICL, 2-shot): SST-2: 76%, AGNews: 79%, DBpedia: 71%, TREC: 35%
>
> Voronov et al. [1] (LLaMA3-Instruct, ICL, 2-shot): SST-2: 72%, AGNews: 61%, DBpedia: 38%, TREC: 29%.
>
> We believe the differences in results arise from two key factors: the difference between the models (LLaMA3 vs. LLaMA3-Instruct) and the high sensitivity of ICL to the choice of the examples the format selection.
> [1] Mind Your Format: Towards Consistent Evaluation of In-Context Learning Improvements, Anton Voronov, Lena Wolf, Max Ryabinin

---

### Official Review · Reviewer_29Ww · 2024-11-08

**Soundness:** 2
**Presentation:** 2
**Contribution:** 2
**Rating:** 3
**Confidence:** 4

**Summary:**

This paper proposes Context-aware Prompt Tuning (CPT) as a new method to address the overfitting problem of prompt tuning. The optimization of CPT consists of a novel loss design and controlled token embedding optimization. Extensive experiments on various datasets using different LLMs demonstrate that CPT achieves better performance than baselines.

**Strengths:**

1. The research problem is interesting, and the authors proposed a new method for better LLM performance on several benchmark classification tasks.
2. The authors have provided detailed literature studies and discussed the motivation of their method,
3. The authors have done extensive experiments to demonstrate the effectiveness of CPT.

**Weaknesses:**

1. The paper is not clearly written and well-organized. It is hard to understand the authors' ideas and the proposed methods. For example, the motivation for presenting Figure 1 and Figure 2 before the introduction is not clear. Although the authors show some input presentation in Sec.3.1, it is still unclear what the inputs look like. The authors should have provided detailed examples in this section.
2. The novelty if this paper is limited, it is a simple combination of several existing works, such as prompt tuning and adversartial training. Besides the empirical results, the paper also lacks theoretical analysis.
3. The figures, such as Figure 2 and Figure 4, are not clear and hard to understand. For example, why are there two x_emb2 in Figure 4. What do different colors mean in these figures?
4.  Some concepts need further clarification. For example, how to do projected gradient descent as presented in Lines 272-273.
5. The authors only work on simple classification tasks and lack experiments on more general generation tasks, such as summarization and machine translation.

**Questions:**

1. Does the final loss have any tuning parameters, as presented in Line 252?
2. See weaknesses.

**Details Of Ethics Concerns:**

N/A.

---

> ### Author Response · Authors · 2024-11-15
> **Answer to 29Ww [1/2]**
>
> Thank you for your thoughtful review and for recognizing the strengths of our work. We are pleased that you found our research problem compelling and appreciated our efforts to improve large language model (LLM) performance through a new method. As you noted, we presented a solid motivation in this work by identifying overfitting as a critical issue in few-shot learning, supporting this claim with empirical evidence, and demonstrating that our proposed method effectively addresses this challenge.
> Our method introduces a novel loss design and applies controlled token embedding optimization, via PGD, to enhance the adaptability of LLMs while mitigating overfitting.
>
> Furthermore, we conducted extensive experiments across various datasets and models to validate CPT's effectiveness. The results consistently demonstrate that CPT outperforms baseline methods. We appreciate your recognition of these strengths, as they underscore the contributions and practical impact of our approach.
> We will now address each of your remarks individually.
>
> "Fig. 2 is not clear."
> We acknowledge that Fig. 2 could benefit from clearer visualization, and we have revised both the figure and its caption accordingly. The confusion likely stemmed from the notation and color choices used to represent the training examples.
>
> The revised Fig. 2 highlights the key distinctions between our method and the baseline approaches, focusing on two primary aspects: the construction of the input (represented with a blue background) and the tokens used during training (represented with an orange background). To improve clarity, we use distinct colors to represent different token types: example tokens are displayed in yellow, while learnable tokens are displayed in pink.
>
> Additionally, the optimization process is visualized by marking tokens involved in loss calculation with red lines beneath them and tokens updated during training with green lines. This detailed representation emphasizes the differences in token utilization and optimization strategies between our method and the baselines.
>
> For example:
> * In In-Context Learning (ICL), we concatenate examples (yellow tokens) without applying any optimization.
> * In Prompt Tuning (PT), we use learnable tokens (pink tokens), and during training we use random training example for loss calculation (indicated by a red line beneath) and updating the learnable tokens (indicated by a green line beneath).
> * In Instruction Prompt Tuning (IPT), both learnable and sample tokens are concatenated, and random training example is used for loss calculation, updates only to the learnable tokens.
>
> In CPT, we initialize the learnable tokens with training examples (similar to ICL). However, unlike ICL, these tokens are optimized throughout training, and the labels of the context contribute to the loss. Therefore, all tokens, including the random training example, are used in loss calculation (indicated by a red line beneath), and all context tokens are updated (indicated by a green line beneath). To signify this dual role, CPT tokens are shown as partially pink and partially yellow, indicating their initialization with training examples and subsequent optimization.
>
>
>
> "why are there two x_emb2 in Figure 4"
> In our method, the training examples serve dual roles — first to construct the context (as in ICL) and then in the optimization step (similar to PT) to update the learnable tokens. However, unlike PT, in our method the context itself consists of these training examples, which are updated during optimization. Detailed example apear in App. G of the revised version.
>
>
>
> "Although the authors show some input presentation in Sec.3.1, it is still unclear what the inputs look like."
> We appreciate this feedback and have added additional clarification on input construction in the updated version (App. G). Specifically, Fig. 4 illustrates how we construct the input when working with two training examples.
> In this process, we first use all training examples to create the context. We then reuse the same training examples to optimize the embeddings of the context tokens. Fig. 4 demonstrates a single optimization step where, in this case, the training example selected for optimization is example number 2.

---

> ### Author Response · Authors · 2024-11-15
> **answer to 29Ww [2/2]**
>
> "The motivation for presenting Figure 1 and Figure 2 before the introduction is not clear."
> We acknowledge that the placement of Fig. 1 and Fig. 2 could be improved. However, these figures are essential to accompany the introduction. Fig. 1 highlights the challenges present in various few-shot learning methods, setting the stage for our work’s motivation. Fig. 2 visually outlines the key differences between existing methods and CPT, providing a clear contrast that helps to frame the problem and our proposed solution early in the paper.
>
>
> "Some concepts need further clarification."
> Thank you for pointing this out, we agree that a detailed explanation of the Projected Gradient Descent (PGD) algorithm is necessary, and we have added it to the updated version(App. H), for clarity. In our approach, we initialize the context tokens, denoted as x_i, using the training examples, with each token x_i associated with a vector delta_i, which is initially set to zero. During the optimization process, the tokens x_i remain fixed, while the delta_i vectors are updated iteratively. After each optimization step, each delta_i is projected to ensure its L2 norm does not exceed a predefined limit, epsilon, thereby controlling the extent of change for each context token.
>
>
> For each token i of the context:
>
>
> \delta_i <- 0
>
>
> x_i <- training_examples_tokens
>
> for j in range(1, num_of_training_steps):
>
>
>    \tab \delta_i =  \delta_i - \alpha (\nabla (Loss(f(x_i+\delta_i), y_i)))
>
>
>    \tab n_i = norm(\delta_i)
>
>
>    \tab \delta_i = \delta_i * (clip(n_i, \epsilon)  / n_i) # projecting delta_i L2 norm
>
>
>
>
> "Does the final loss have any tuning parameters, as presented in Line 252?"
> No, there are not additional paremeters, this is the exact expression that is used.
>
>
> "lacks theoretical analysis."
> We acknowledge that the paper lacks a formal theoretical analysis. However, as you stated, we supply “Extensive experiments on various datasets using different LLMs”. We have also provided an empirical analysis, such as the loss graph in Fig. 1, which supporting our motivation.
>
>
> "Lack of novelty"
> While we recognize that our approach builds upon established methods, our contributions lie in the novel integration of In-Context Learning (ICL), Prompt Tuning (PT), and adversarial strategies within a cohesive framework, Context-Aware Prompt Tuning (CPT). As you noted, we introduce a novel loss design that incorporates context labels alongside the standard loss, while weighting each token with a decay factor to leverage the "recency bias".
> Additionally, using PGD, our optimization process features a unique approach to controlling token embedding modifications ensuring updates remain within a defined range. This combination enables CPT to effectively address overfitting and improve few-shot classification performance, as demonstrated through extensive empirical results. By unifying these elements, CPT offers a distinct approach to few-shot learning, achieving significant performance gains over baseline methods. We appreciate your recognition of these novel components, which are central to our contributions.
>
>
>
> "lack experiments on more general generation tasks"
> Our work focuses specifically on enhancing few-shot classification tasks, rather than general generation tasks, and we did not claim that our method is universally applicable across different task types. As you noted, we conducted extensive experiments on various datasets using different LLMs, all within the classification domain, to support this focus. Our evaluations included five datasets — one we created and four publicly available ones — and tested our method on three different large language models, consistently demonstrating CPT’s effectiveness in few-shot classification settings.

---

### Author Response · Authors · 2024-11-20
**Discussion Period**

We have provided detailed comments addressing each of the reviewers' points, and to fully benefit from the discussion period, we would kindly like to remind the reviewers that we are awaiting their responses during this time. Your feedback is invaluable in ensuring the clarity and quality of the final version, and we greatly appreciate your continued engagement in this process.

---

### Author Response · Authors · 2024-11-20
**All Reviewers [1/2]**

We would like to extend our deepest gratitude to all the reviewers for their time and constructive feedback. Your insights have been included in our paper, and we truly appreciate your contributions. Throughout this response, we will refer to reviewers 29Ww, CoLs, DESR, and iktw as R1, R2, R3, and R4, respectively.


You described this as an “interesting research problem” (R1) and acknowledged that we have “proposed a new method for better LLM performance on several benchmark classification tasks” (R1). Additionally, you noted that we propose an “extension of prompt tuning by combining it with in-context learning” (R2) and highlighted that our “method is straightforward and easy to understand” (R3).
The clarity of our presentation was further commended, as you mentioned that “the figures (Fig. 2) in the paper are really helpful in understanding key differences between their method and various baselines they implemented” (R2). You also pointed out that we “provided detailed literature studies and discussed the motivation of their method” (R1), particularly in revealing and addressing the overfitting problem.


Moreover, you recognized the novelty of our approach, stating that “the idea of optimizing the context using PGD is novel” (R4) and that “it makes sense to use PGD so that the examples will not change by too much and the optimization does not overfit” (R4).


Finally, you acknowledged the robustness of our evaluation, emphasizing that “the authors have done extensive experiments to demonstrate the effectiveness of CPT” (R1). You described the experiments as “comprehensive, using three models and testing on several classification datasets” (R3) and noted that our work evaluates “the model on different open-source datasets like AGNews, SST-2, DBpedia, and TREC” (R2). Importantly, you highlighted that “the proposed method outperforms LoRA, PT, IPT, and ICL in most cases” (R3).


As for the points that required further elaboration, we have responded to each of you individually, addressing all of your concerns thoroughly. Additionally, we would like to highlight your remarks that necessitated modifications to the paper or were shared among some of you. For your convenience, all changes made to the paper have been marked in red in the revised version.





(R1) “Fig. 2 and Fig. 4 are not clear” - We revised Fig. 2 and added additional explanations for Fig. 4 to improve clarity (in App. G).


(R1, R4) “it is still unclear what the inputs look like” and “It is unclear what is the difference between PT \dag and IPT \dag.” - This was addressed in App. G.


(R1) “Some concepts need further clarification” - We included a detailed explanation of how PGD works to provide further clarity (in App. H).


(R2,R3) “abstract of this paper is hard to understand” and "Why ICL cannot fully extract the information that exists in the training examples" - We updated the abstract.

(R2, R4) "description provided on the set classification" and "Set Classification dataset is confusing" - As requested by R4, we have removed the Set Classification dataset from the paper.



(R4) “There is one possibility that the performance should be attributed to the use of PGD” - We added additional experiments (in App. I), as requested, to confirm that the improvements are not solely due to PGD. Additionally, we emphasized that the ablation of the original version (Tab. 2) already demonstrates the critical role of the novel loss function in our method.

---

### Meta-Review · Area_Chair_b2x4 · 2024-12-20

**Metareview:**

This paper introduces Context-aware Prompt Tuning (CPT), a method combining In-Context Learning (ICL), Prompt Tuning (PT), and adversarial optimization to address overfitting in few-shot learning tasks. The approach is validated on multiple classification datasets and different LLMs, demonstrating improved performance over baselines. However, concerns about the novelty, dataset relevance, and generalization to broader tasks were widely raised among the reviewers, and the authors, though actively engaging in the discussion, do not fully address the reviewers' concerns.

I think this work has its methodological merits and practical significance, but it needs a revision to better illustrate these points to the readers. So given that most reviewers find these problems are not fully unaddressed in the current form, I recommend rejection but recommend the authors to submit it to a future venue taking these comments into consideration.

**Additional Comments On Reviewer Discussion:**

Unfortunately, despite the authors prepared a detailed response, not all reviewers have responded or at least acknowledged the rebuttal, even after reminders sent through AC. Reviewer DESR has increased the score to 5 after the rebuttal, but still leans towards rejection. Even  Reviewer DESR gives a score of 6, they also agree on the lack of enough novelty in the presentation. Therefore, I believe that is a consensus and thus leads to my recommendation for rejection.

---

### Decision · Program_Chairs · 2025-01-22

Reject